# Contextual and Psychosocial Factors Influencing the Use of Safe Water Sources: A Case of Madeya Village, uMkhanyakude District, South Africa

**DOI:** 10.3390/ijerph17041349

**Published:** 2020-02-19

**Authors:** Chanelle Mulopo, Chester Kalinda, Moses J. Chimbari

**Affiliations:** 1School of Nursing and Public Health, College of Health Sciences, University of KwaZulu-Natal, Howard Campus, Durban 4001, South Africa; ckalinda@unam.na; 2Faculty of Agriculture and Natural resources, University of Namibia, Katima Mulilo Campus, Winela Road, Box 1096, Ngweze, Katima Mulilo, Namibia; 3School of Nursing and Public Health, College of Health sciences, University of KwaZulu-Natal, Howard Campus, Durban 4001, South Africa; chimbari@ukzn.ac.za

**Keywords:** RANAS model, water, behaviour, psychosocial factors, rural

## Abstract

*Background*: Schistosomiasis is a public health problem that affects over 240 million people worldwide. It is mostly prevalent in tropical and sub-tropical areas among communities with limited access to clean water and adequate sanitation. This study was conducted in uMkhanyakude District in rural South Africa, where water resources are limited. In this community, individuals frequently come into contact with freshwater bodies for various reasons. The objective of the study was to identify critical contextual and psychosocial factors for behaviour change to reduce risk of schistosomiasis transmission in Madeya Village, uMkhanyakude district. *Methods*: Structured household interviews were held with 57 primary caregivers to assess their thoughts and attitudes towards collecting water from a safe source. We used the Risk, Attitude, Norm, Ability, and Self-regulation model (RANAS) to estimate the intervention potential for each factor by analysing differences in means between groups of current performers and nonperformers who use safe water sources. *Results*: The subscale vulnerability belonging to the risk factor on the RANAS was scored as low. Furthermore, attitudinal factors towards the use of safe water sources were found to be low. Ability factors (confidence in performance and confidence in recovery) towards the use of safe water sources were low as well, indicating that these factors should be the target of the intervention in the study area. *Discussion*: Based on this study, it is recommended that a community-based empowerment intervention strategy it appropriate. The strategy should prompt behavioural practice and public commitment, use persuasive language to boost self-efficacy and target younger low-income caregivers between 18 and 35 years of age.

## 1. Background

Approximately 430 million people worldwide use unimproved water sources and 144 million people use surface water [1]. Lack of access to improved water sources contributes to the high burden of diseases such as diarrhoea and schistosomiasis [2]. Schistosomiasis is a public health problem which affects over 240 million people worldwide with 500 million people at risk of infection [3,4]. Over 90% of cases of schistosomiasis reported are in sub-Saharan Africa [5]. It is mostly prevalent in tropical and sub-tropical areas among communities with limited access to clean water and adequate sanitation [6]. Transmission of schistosomiasis is dependent on the presence of the intermediate host snail (IHS) and human contact with cercariae infested waters [7]. Since infection occurs during human contact with water, it can be avoided through provision of safe water sources. The use of safe water sources is however dependent on sound understanding of psychosocial drivers of behaviour.

Understanding psychosocial factors is crucial for achieving a desired behaviour that can be sustained over time. There are a number of factors referred to as behavioural determinants, which influence the adoption of a specific behaviour and continuation of practicing the behaviour [8]. For successful and sustained behavioural change, the changes need to occur in psychosocial factors such as attitudes, norms and belief systems that determine behaviour [8,9,10].

Although schistosomiasis is prevalent in uMkhanyakude [11,12,13], to our knowledge, no structured assessment of psychological determinants that influence the use of safe water sources as a means to reduce risk for schistosomiasis transmission has been conducted. Our study sought to identify the psychosocial factors influencing the use of safe water sources in order to develop a behavioural change intervention for reducing schistosomiasis transmission in Madeya Village in uMkhanyakude district. We used the Risk, Attitude, Norms, Ability and Self-regulation (RANAS) model to identify psychosocial factors that influence the use of safe water sources in Madeya. The RANAS model was developed for prediction of health behaviours in developing countries and is based on psychological health behaviour [10]. The model is an integration of psychological factors proposed by theories of behavioural change such as the health belief model, the health action process approach and the theory of planned behaviour [8]. Additionally, the model also provides behavioural change techniques that tackles the factors that need to be changed. The model depicts five distinct components or “factor blocks” that have to be favourable for the new behaviour to be assimilated. These include risk factors, attitudinal factors, norm factors, ability factors and self-regulations. The first block comprises the risk factors which refers to the person’s understanding and/or awareness of the health risk associated with contracting the disease. The second block is the attitudinal factors that refer to a person’s negative or positive feeling towards a particular behaviour. Normative factors which form the third block represent the perceived social pressure to engage in a specific behaviour. Ability factors constitute the fourth block and represents an individual’s perceived confidence in their ability to perform a specific behaviour. Self-regulation, which forms the last block, represent a person’s attempts to plan and self-monitor a behaviour and to manage conflicting goals and distracting cues [14,15]. We sought to address the contextual and psychosocial factors influencing the use of safe water sources in Madeya. Our findings may be used to design an intervention framework that can be applied to control schistosomiasis transmission in Madeya.

## 2. Materials and Methods

### 2.1. Study Area and Context

The study was performed in Madeya village, located in uMkhanyakude district of KwaZulu-Natal province, South Africa (Figure 1). Madeya is just one of the sites where the large study is being conducted. UMkhanyakude district is one of the poorest districts in KwaZulu-Natal and much of it is characterized by poor economic development. Most of the rural areas within the district are poor, have limited infrastructure and experience poor service delivery [16]. Water supply is poor and sanitation facilities are inadequate, thus creating a conducive environment for transmission of infectious diseases such as schistosomiasis and soil transmitted helminths [17,18].

### 2.2. Data Collection Methods, Study Participants and Inclusion Criteria

A standardized questionnaire was administered face-to-face to 57 household heads selected randomly using the modified random-route procedure which involves dropping off interviewers at different locations within the designed geographic coverage area and letting them choose a starting point and direction for selecting households [19]. Only households with at least one child under the age of five years were selected to participate in the study. As a result, 57 households were selected from a pool of 165 who are part of the larger study.

The questionnaire was designed in English and translated to isiZulu, the local language used in the study area. Local research assistants who administered the questionnaire received intensive training over two days. To ensure uniformity in the understanding and evaluation of the data collection, the instrument was pre-tested in the Mgedula Village, which has similar socio-demographics and cultural characteristics to the study area. Further, modifications were made to the tool based on findings from pre-testing.

The questionnaire included questions on demographics such as age, gender and number of people in the households. To determine the use of safe water sources, respondents were asked to identify their main source of water. Several items were constructed for each psychological factor to address all the RANAS components in detail. The questions were arranged in a logical sequence and uploaded on KoboCollect, an online open source platform for collecting and analysing data [20]. The study took place between November 2017 and November 2018.

Ethical approval was obtained from the Humanities and Social Science Research Ethics Committee at the University of KwaZulu-Natal. Protocol reference number: HSS/0396/018D. All participants gave informed consent to participate in the study.

The RANAS model was the most applicable model because, contrary to other behavioural change models, the RANAS measures more behavioural factors [21]. Additionally, other approaches rely on provision of knowledge as a tool to change behaviour, yet psychological theories inform us that behavioural knowledge alone is not sufficient to bring about behavioural change [21]. The RANAS model presents a systematic approach of behavioural change informed by psychological theory and behavioural interventions [8].

### 2.3. Data Analysis

The incidence of each factor targeted for change was measured and analysed for its intervention potential. To determine the intervention potential (IP) for each factor, the improvement reserve was first analysed and then its impact on behaviour. The improvement reserve (IR) is defined as the difference between the population mean (Mean) in a factor and the maximum possible value (Max) of this factor (IR = Max − Mean) [8]. To determine the impact of each factor compared to other factors on behaviour, regression analysis was calculated. In the regression analysis, the B-values express how the size of each factor influences the behaviour compared to the others, provided that the factors are transformed to the same range of values. Finally, to fix the behaviour improvement potential (IP) of each factor, the IR was multiplied by the B-value derived from the regression analysis (IP = IR × B). Furthermore, to determine what psychosocial factors are important behavioural drivers for the use of safe water sources in the study area, the intervention potential (IP) for each psychological factor of the five RANAS model components was estimated by comparing the means between groups of performers, (doers) individuals who collect water from safe water sources and nonperformers, (non-doers) individuals who collect water from unsafe water sources [8]. Total population means as well as differences in means (*t*-tests) between the groups of doers and non-doers were calculated for all psychological variables. The Guideline for Behaviour Change stipulates that the IP for psychological factor components and their subscales results from the distance of mean value to scale maximum [22]. In our study, all factors falling at or below the mid 3-point value on a scale of 1–5 were considered important for the design of the intervention, because of the remaining theoretical improvement reserve.

We therefore calculated the IP as a combination of the distance of the total mean from the desired scale maximum value (5-point scale end) as well as the difference between the group means [8]. T-tests were, run to confirm statistically significant differences of means between the groups of doers and non-doers for all five of the RANAS components. Values between 1 and 2 are considered as a low, 2–3 as a moderate, 3–4 as a high, and above 4 as a very high IP. Since questions about self-regulation cannot be answered by non-doers and because we had a small sample for doers, we did not measure the self-regulatory component. All analyses were performed using SPSS version 25.

## 3. Results

### Study Population

All the targeted 57 households consented to participate in the study. The mean age of the respondents was 37 years (standard deviation [SD] = 13.23). Most households (43.9%, *n* = 25) had 6–10 members. The majority (66.6%, *n* = 38) of the households had one child below the age of five. Most (29.8%, *n* = 17) of the participants earned ZAR201–500 (USD13.53–33.65) per month (1USD = ZAR14.86, https://www.bloomberg.com/quote/USDZAR:CUR, date accessed on 14 February 2020). Table 1 presents results grouped into two categories: doers and non-doers. Doers are individuals who use safe water sources at least 60% of the time and non-doers are individuals who mostly use unsafe water sources.

There were 8.8% (*n* = 5) male and 91.2% (*n* = 52) females. Half (50.9%; *n* = 29) of the participants were not married. Close to half of the participants (42.1%, *n* = 24) had completed secondary school education and more than half the participants (77.2%, *n* = 44) were unemployed. Most of the participants (31.6%, *n* = 18) resided in dwellings made of traditional mud block and corrugated iron.

We asked the respondents where they collected water in order to identify the proportion of the population that had access to improved water sources and the proportion that had access to unimproved water sources. Bearing in mind that the proportion of the sample that had access to unimproved water sources were at a greater risk of contracting schistosomiasis. Primary water sources used in the supply of drinking water included the tap (10.53%) and pump/bore hole (7.02%), and the majority of the respondents collected water from freshwater bodies such as unprotected dug wells, rivers and dams [surface water (77.19%)]. The municipality also provided water using a water tanker (3.51%) (Table 2).

In order to determine the psychosocial factors that influence where people collect water, the position of total means as well as differences in means on the standard scale were used to estimate the IPs for all psychological components of the RANAS model. Figure 2 shows the outcome of all psychosocial factors. For the targeted behaviour, perceived vulnerability was measured by asking how high or low the risk for contracting schistosomiasis from freshwater bodies was (scale range for all questionnaire items were rated from 1 = very low to 5 = very high). On average, perceived vulnerability was rated very low (M = 0; SD = 3.485). There was a moderate difference between doers and non-doers of 0.063 scale points, indicating no statistically significances between the two groups (*t* = 0.198, *p* = 0.422). Extent of the schistosomiasis problem was measured by asking how severe the respondents rated schistosomiasis. The average perceived severity was rated moderate (M = 2.543; SD = 1.637). There was a significant difference (*t* = −2.065, *p* = 0.021) of 1.15 scale points between doers and non-doers. The IP for the subscale vulnerability was high whereas that of severity was moderate but when combined, the subscales produced a moderate IP.

To measure attitudes, feelings towards collecting water from a safe source were measured by asking the respondents how they felt about collecting water from a safe source. The outcome was moderate (M = 2.286; SD = 3.03) with a large difference of 2.04 scale points between the two groups of doers and non-doers. The difference (*t* = −1.978, *p* = 0.026) between doers and non-doers. The mean score for cost benefit was low (M = 1.825; SD = 2.399) with a moderate insignificant difference (*t* = −0.905, *p* = 0.184) between doers and non-doers of 0.76 scale points. Both attitudinal subscale scores were low, indicating that respondents had negative feelings towards collecting water from a safe source and found it very time consuming and effortful. The IP for all factors within the component attitude were therefore high.

Concerning norms, the perception of how common the behaviour of collecting water from a safe source was observed in the community was scored low (descriptive norm; M = 1.578; SD = 1.413). The difference between doers and non-doers was moderate, up to 1.0 scale point. There was a significant difference between groups of doers and non-doers (*t* = 2.082, *p* = 0.021). Furthermore, injunctive norms which represent people’s experience of how strongly they felt, people in authority or their leaders promote the use of safe water sources was also rated low (M = 1.439; SD = 2.946). The difference between doers and non-doers was also large (91.02 scale points) but not significantly different between doers and non-doers. (*t* = −0.991, *p* = 0.162). The two subscales resulted in a moderate IP for the norm component.

Ability factors that measure people’s perceptions of their own skills to pick up and maintain the behaviour of collecting water from a safe source were low (M = 1.771; SD = 1.721). The difference between groups was large at 1.01 scale points and it was statistically significant (*t* = 1.703, *p* = 0.04). Furthermore, the subscale that measures recovery from a drawback (self-efficacy) was rated very low (M = 0.561; SD = 3.245). An insignificant difference (*t* = 1.338, *p* = 0.09) of 1.51 scale points between the doers and non-doers was observed. The combined effect of confidence in performance and self-efficacy translated to a high IP.

Self-regulation factors were not assessed because we had a small sample of doers which did not permit assessment of self-regulation among non-doers.

## 4. Discussion

The psychosocial data provide an evidence-based choice for appropriate Behavioural Change Techniques (BCTs) for promotion of use of safe water sources. We used an innovative approach that contributes to the knowledge of schistosomiasis control. This approach contributes strongly to preventative approaches, which fall within the theme of health promotion. While the mainstay of schistosomiasis control focuses on treatment of the disease, the health promotion approach focuses on preventing disease thus promoting health. We used the RANAS model to identify psychosocial factors that determine the use of safe water sources in Madeya village, located in uMkhanyakude district. These psychosocial factors inform an intervention framework for protecting community members from contracting schistosomiasis. Schistosomiasis can be prevented when these factors are targeted and changed, resulting in more people using safe water sources and avoiding unsafe water sources.

Our research findings show that perceived vulnerability was rated low and the difference between groups of doers and non-doers was not significant. This means that both groups did not perceive themselves as vulnerable to schistosomiasis and diarrhoea. Furthermore, severity was scored moderate but with a significant the difference in scores between groups of doers and non-doers indicating that doers did perceive schistosomiasis and diarrhoea to be serious diseases while non doers did not. A study conducted in Burundi reported that an increase in knowledge about diarrhoea diseases resulted in a lower prevalence of the disease [23]. Therefore, health education or hygiene education is critical if one aims to increase perceived vulnerability and severity of a disease. Our findings also show that the majority of the community relied on surface water as the main source of water. Results from a study conducted in Ghana reported that inadequate access to potable water was associated with a high incidence of diarrhoea [24]. Furthermore, the study suggested that childhood diarrhoea decreased with availability of standpipes and private indoor pipes. In our study we found that, only a few households had access to piped or pumped water sources. Low perceived vulnerability and severity of the disease may be a contributing factor to the use of surface water or unsafe water sources.

Our results show that for the risk component, the intervention potential for the sub-component vulnerability was high and that of severity was moderate. The combined subscales produced a moderate IP for the risk component. The intervention needs to target the perceived vulnerability psychosocial factor because the results are informing us that respondents do not perceive themselves as vulnerable to schistosomiasis even though schistosomiasis is prevalent in the study area [18]. The risk component is therefore considered important for the design of intervention programmes. A previous study on knowledge attitude and practice (KAP) on schistosomiasis in the study area reported poor knowledge about schistosomiasis [25]. This could be the reason why there is low perceived risk of schistosomiasis among members of the community even though the disease is prevalent. These results are similar to a study conducted in Chad that reported low perceived risk of cholera despite the high prevalence of the disease in the area because no Cholera cases were observed prior to the survey [10].

Given that infected snails have been found at some of the water source points used by the community in this schistosomiasis endemic area, there is need for an intensified health promotion campaign to change people’s behaviour and perceptions regarding schistosomiasis. There are many strategies that can be used to bring about changes in behaviour. Dreibelbis et al. [26] reported environmental nudges to have successful results in promoting handwashing behaviour. Similarly, environmental nudges can also be applied in this community to prompt members of the community to use safe water sources.

For the attitude component, respondents scored moderate towards how they felt about using safe water sources and scored low on the cost–benefit scale. This means that the majority of the community members had negative attitudes towards the use of safe water sources. They felt that collecting water from a safe source needed more effort and was time consuming. One of the reasons for this observation was the limited number of improved water sources in the study area [27] which resulted in people having to wait for a long time in queues or having to travel long distance to collect water from a safe sources. Limited resources in the environment (contextual factors) resulted in negative attitudes towards collecting water from safe sources as indicated by the low scores on the cost–benefit scale for both doers and non-doers. Limited resources had a time cost to it as perceived by members of the community, meaning that individuals had to spend more time collecting water from improved sources than they would if they collected water from the rivers. Additionally, they had to be at public taps at specific times, usually at odd hours such as very early in the morning, because that would be the only time that water would be available. Other studies have indicated that improving access to clean water coupled with hygiene education can bring about behavioural change [28,29,30].

The attitude component produced a high IP. The high IP for attitude component requires that persuasive messages be included in the intervention to improve attitudes towards collecting water from a safe source. The message should highlight the health benefits of collecting water from safe sources as well as the health consequences of collecting water from unsafe water sources.

Both subscales on the normative factor (descriptive and injunctive norms) were scored low. We found a significant difference between doers and non-doers on the descriptive norm scale. The low subscale scores mean that majority of the community members were not collecting water from safe water sources. In other words, the behaviour was not commonly practiced. In order to change behaviour, one needs to engage respected individuals in the community to ensure that they adopt the behaviour and that will have an influence on other members of the community who would want to model the behaviour. In a similar study conducted in Bolivia where the authors wanted to increase the adoption of a water technology, their results showed that the influence of opinion leaders had desirable outcomes during the middle of the diffusion process [31].

The normative component resulted in a moderate IP. Moreover, there was a significant difference between doers and non-doers on the descriptive norm scale with the non-doers scoring much lower than the doer group. This means the use of safe water sources was not commonly observed among non-doers compared to the doers. Therefore, the recommended behavioural change technique should prompt public commitment to collect water from a safe source as well as have the support of community leaders to promote the use of improved water sources to promote injunctive norms.

Confidence in performance and self-efficacy as subscales of the Ability component were both scored low. This means that members of the community did not feel confident in their ability to consistently use improved water sources. Moreover, this can also be attributed to the limited improved water sources in the community. Studies in West Africa that have looked at the influence of psychosocial factors in the prevention of Ebola and found that the ability component was an important factor in the prevention of Ebola [14,32].

Abilities factors (confidence in performance and self-efficacy) revealed a high IP. This means the intervention should target ability factors. We found that there was a significant difference between groups of doers and non-doers on the scale of confidence in performance. Non-doers rated themselves as less confident in having the skill to use and maintain the use of safe water sources compared to the doers. This outcome can also be attributed to limited improved water sources in the community that could have made people to believe that it was difficult to develop a habit of using safe water sources. The behavioural change technique recommended for this psychosocial factor is the provision of safe water sources such as boreholes and communal taps as well as prompt coping and recovery from a relapse. This includes telling people that lapses are normal when adopting a new behaviour.

Our findings show that the Madeya community relies on surface water (unprotected dug wells, rivers and dams) as the main source of water for domestic use. This implies that the risk of contracting schistosomiasis infection from exposure to cercariae infested waters was high [33,34]. Previous studies in the same area reported high prevalence of schistosomiasis among school-going children [11,12].

## 5. Conclusions

We determined psychosocial factors that influence people’s behaviours with regards to the use of safe water sources. The factors that resulted in a high intervention potential are vulnerability, attitudinal and ability factors. Therefore, the Water, Sanitation and Hygiene (WASH) intervention needs to focus on these psychosocial factors in order to increase the use of safe water sources. Based on these findings, a community-based empowerment intervention strategy is recommended to prompt change in behavioural practice and public commitment. Furthermore, use of persuasive language to booster self-efficacy is indicated. The intervention should be targeted to younger caregivers with low income as the older age group and high-income groups seemed to do better than the former groups.

## Figures and Tables

**Figure 1 ijerph-17-01349-f001:**
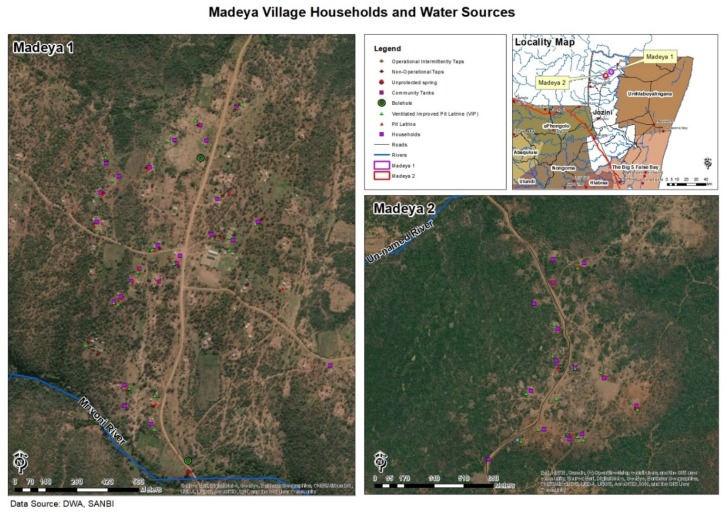
Aerial and coordinate map of the households included in the Madeya Village, the study area.

**Figure 2 ijerph-17-01349-f002:**
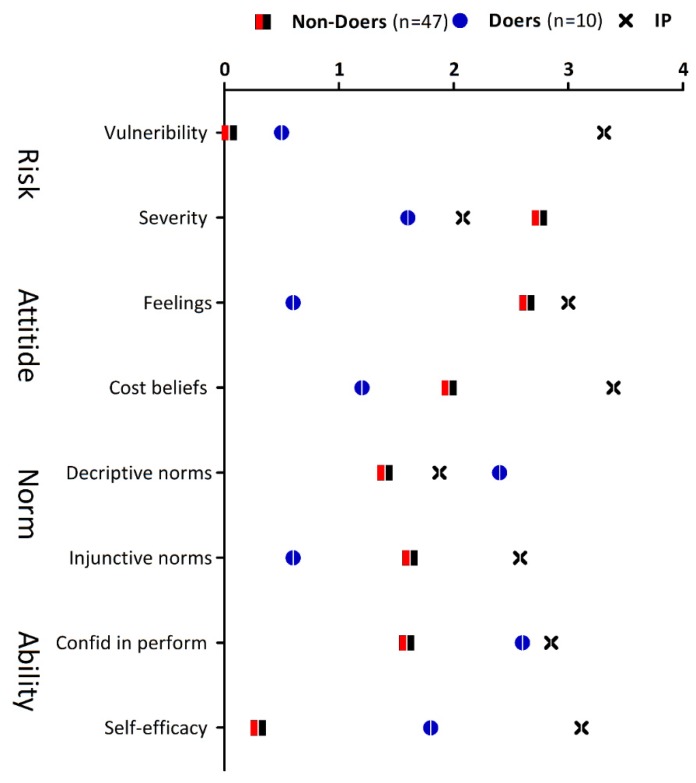
Results of the Risk, Attitude, Norm, Ability, and Self-regulation analysis, with a focus on risk and ability.

**Table 1 ijerph-17-01349-t001:** Characteristics of the study participants separated into two groups: doers and non-doers.

Percentage		
	Doer (*n* = 10)	Non-Doer (*n* = 47)
Age		
	43.70 ± 16.51	35.85 ± 12.22
Marital status		
Married	10	19.15
Divorced	0	4.26
Single	30	55.32
Widowed	20	2.13
Cohabiting	40	19.15
Highest level of school completed		
No school completed	40	19.15
Primary school	30	29.79
Secondary school	30	44.63
College level	0	2.13
University	0	4.26
Occupation		
Student	0	4.26
Self-employed	10	10.65
Employed by Government	0	2.13
Employed by private company	0	4.26
Unemployed	90	74.47
Retired	0	4.26
Type of dwelling		
Mainly traditional mud block with thatch	10	17.02
Mainly traditional mud block with corrugated iron	30	31.91
Mud plastered with cement	0	21.28
Concrete or cinder block	30	21.28
Modern brick house	30	4.28
Temporary structure	0	0
Other	0	4.28

**Table 2 ijerph-17-01349-t002:** Primary sources of water in the Madeya village.

Primary Source of Water	*n*	%
Tap	6	10.53
Pump	4	7.02
Surface	44	77.19
Tanker	2	3.51
Not classified	1	1.75
Total	57	100

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
