# Peer review of "Contextual and Psychosocial Factors Influencing the Use of Safe Water Sources: A Case of Madeya Village, uMkhanyakude District, South Africa"

_ijerph, 2020, doi:10.3390/ijerph17041349_

Round 1

Reviewer 1 Report

Dear authors, thank your for your cover letter addressing my previous comments. I believe that the article is still in lack of a proper map that could provide the context of the case study. I could not reach the Appendix you mention in the cover letter. I believe it is not available in the web plattaform. Nevertheless my suggestion is to include it as a figure in the article. Since this could be a matter of Journal editing I submit this suggestion to editors and to your consideration.  

Apart from that, I have the following minor suggestions to your manuscript:

Line 23: “We used the the Risk”. ‘The’ is repited.

Line 81: “province, South Africa as part of a larger. Madeya” ‘Of a larger’ what?

Lines 126-129 There is no need to repit explanations for doers and non-doers in these sentences so closed to each others.

Line 149: “There were 8.8 % (n=5) male and 91.2% (n=52) females. Fifty point 9 percent (50.9%; n = 29)”. “Fifty point nine” or better change it to figures as the beginning of the sentence.

Line 318: “public commitment. Furthermore, use of persuasive language to booster self-efficacy is indicated The” Full stop is missing.

Author Response

Reviewers comments

Author’s response

Pages where changes are made

Reviewer 1

Include the map as a figure in the article

We have inserted the map of the study area as well as the household coordinates

Materials and Methods: Figure 1 Page 3

Line 23: “We used the the Risk”. ‘The’ is repeated.

We have removed the repetitive “the”

Abstract: line 23 page 1

Line 81: “province, South Africa as part of a larger. Madeya” ‘Of a larger’ what?

We have removed “as part of a larger”.  It was meant to be deleted.

Materials and Methods: line 78 page 2

Lines 126-129 There is no need to repeat explanations for doers and non-doers in these sentences so closed to each others.

We removed the second explanation.

Materials and Methods: line 126 page 4

Line 149: “There were 8.8 % (n=5) male and 91.2% (n=52) females. Fifty point 9 percent (50.9%; n = 29)”. “Fifty point nine” or better change it to figures as the beginning of the sentence.

We have removed the wording and kept the numbers

Materials and Methods: line 148 Page 4

Line 318: “public commitment. Furthermore, use of persuasive language to booster self-efficacy is indicated The” Full stop is missing.

We have included the “full stop”.

Conclusion: line 313 page 9

Reviewer 2 Report

The authors have signicantly improved the quality of the manuscript according with reviewers suggestions. 

Author Response

There are no comments provided by the reviewer.

This manuscript is a resubmission of an earlier submission. The following is a list of the peer review reports and author responses from that submission.

Round 1

Reviewer 1 Report

Dear authors,

I find the subject of your manuscript within the aims and scope of the Journal. It is an original and interesting contribution to the field of factors influencing people behavior regarding the use of drinking water.

However, from my point of view, the paper requires some improvements to be more scientifically clear. I consider that more information about the methodology, specifically about the model and five distinct components or “factor blocks’ would be very positive as well about the intervention potential (IP) concept. Even I would suggest including an explanation about the rationale behind these factors. Line 60 of the papers says: “The model depicts five distinct components or “factor blocks”. But you don’t even mention them.

I also consider that it would help if you contextualize more the situation of this case study. Is it possible to provide a map of the dwellings and the water sources? What about the distance to fetch the water, is it taking into account in this model?

Furthermore, could you compare these results with other countries’ or regions’. Are there similar studies in other countries? Is this an innovative approach? I think you could add information about it in the introduction and the conclusion sections in order to make it more general and more appealing for more readers.

Apart from that please be aware that table 4 is missing in this manuscript version: Line 135-136: Table 4 describes the outcome of all 
psychosocial factors.

I hope you find my comments useful.

Reviewer 2 Report

The aim of the paper was identify the psychosocial factors influencing the use of safe water sources in order to develop a behavioral change intervention to reduce schistosomiasis in Madeya Village in uMkhanyakude district using RANAS (Risk, Attitude, Norms, Ability and Self-regulation) model. The topic is of relevance considering that schistosomiasis is a public health problem that affects almost 240 million people worldwide, namely tropical and sub-tropical areas among communities with limited access to clean water and adequate sanitation. It is suggested to the authors:

3.1. It is suggested a professional English revision of the manuscript.

3.2. At Table 1 (line 123):

            3.2.1. The results should be presented uniformly with the same decimal places.

            3.2.2. The age should be presented ± SD (standard deviation) in both groups.

            3.2.3. The total number of doers and non-doers should be identified.

            3.2.4. What means (-)? 0 (zero)? Is should be clearly stated.

3.3. At Table 2 (Table 124) it is presented the frequency (%) but it is not correctly identified.

3.4. Only 3 references were used to discuss all manuscript. The discussion should be deeply reformulated in order to better discuss the obtained results with the available literature.

3.5. The sentence should be reviewed (lines 251-252).